# Phosphate-Solubilizing Bacteria with Low-Solubility Fertilizer Improve Soil P Availability and Yield of Kikuyu Grass

**DOI:** 10.3390/microorganisms11071748

**Published:** 2023-07-04

**Authors:** Daniel Torres-Cuesta, Duber Mora-Motta, Juan P. Chavarro-Bermeo, Andres Olaya-Montes, Cesar Vargas-Garcia, Ruth Bonilla, German Estrada-Bonilla

**Affiliations:** 1Corporación Colombiana de Investigación Agropecuaria (AGROSAVIA)—Tibaitatá, km 14 via Mosquera, Mosquera 250047, Colombia; dtorres@agrosavia.co (D.T.-C.); jchavarro@agrosavia.co (J.P.C.-B.); cavargas@agrosavia.co (C.V.-G.); rbonilla@agrosavia.co (R.B.); 2Centro de Investigaciones Amazónicas Cimaz-Macagual, Universidad de la Amazonia, Florencia 180002, Colombia; du.mora@udla.edu.co; 3Departamento de Ciencias do Solo, Universidade Federal de Lavras, Lavras 37200-000, Brazil; an.olaya@udla.edu.co

**Keywords:** diammonium phosphate, rock phosphate, compost, microbial inoculants, PGPR, phosphate sequential fractionation, enzymatic activity

## Abstract

Inoculation with phosphate-solubilizing bacteria (PSB) and the application of phosphorus (P) sources can improve soil P availability, enhancing the sustainability and efficiency of agricultural systems. The implementation of this technology in perennial grasses, such as Kikuyu grass, for cattle feed in soils with high P retention, such as Andisols, has been little explored. The objective of this study was to evaluate the productive response of Kikuyu grass and soil P dynamics to BSF inoculation with different P sources. The experiment was conducted on a Kikuyu pasture, which was evaluated for 18 months (September 2020 to March 2022). Three P fertilizers with different solubility levels were applied: diammonium phosphate (DAP) (high-solubility), rock phosphate (RP), and compost (OM) (low-solubility). Moreover, the inoculation of a PSB consortium (*Azospirillum brasilense* D7, *Rhizobium leguminosarum* T88 and *Herbaspirillum* sp. AP21) was tested. Inoculation with PSB and fertilization with rock phosphate (RP) increased soil labile P and acid phosphomonoesterase activity. Increased grass yield and quality were related with higher soil inorganic P (Pi) availability. This study validated, under field conditions, the benefits of PSB inoculation for soil P availability and Kikuyu grass productivity.

## 1. Introduction

Kikuyu grass (*Cenchrus clandestinus* [Hochst. ex Chiov] Morrone) is the most important perennial grass in the high tropics of Colombia [1,2]. This crop exhibits adequate development under optimal fertilization, increasing biomass availability and promoting higher milk production [3,4]. In Colombia’s high tropics, Kikuyu grass requires large amounts of phosphate fertilizers to adequately grow and maintain high nutritional quality [5,6]. Commonly, this grass is established on Andisol, whose main characteristic is high phophate (P) adsorption (75–90%) on metal–cation complexes (calcium phosphates or aluminum/iron phosphates). The mineralogy of these soils is dominated by amorphous or low-order crystalline clay minerals (allophane, imogolite, and ferrihydrite) [7,8]; however, these minerals can be immobilized by functional groups of humic substances in the soil organic fraction [9] or lost by leaching [10]. Many Andisols contain high concentrations of total P, of which only a very low fraction is available for plants [11,12,13,14,15]. Thus, there is an imbalance between soil P supply and plant nutrition [16], with large amounts of unavailable P accumulating in the soil [13,14].

Consequently, efforts have been made to develop technological strategies to promote adequate amounts of bioavailable P in soils and to improve P uptake by plants. One of these strategies is the use of phosphate-solubilizing microorganisms as bioinoculants in agriculture [17]. Within this group, phosphate-solubilizing bacteria (PSB) have been shown to potentially promote the growth of commercially important crops, such as sugarcane [18], cotton [19], and grazing forage species [20]. In addition, they have demonstrated the ability to modify P lability, increasing its soil availability [21,22,23].

The PSB strains evaluated in this study were isolated from Kikuyu grass (*Herbaspirillum* sp. AP21 and *Azospirillum brasilense* D7) and active red clover (*Trifolium pratense*) nodules (*Rhizobium leguminosarum* T88). These strains were selected because previous studies have demonstrated their ability to promote growth in associated ryegrass and red clover pastures under abiotic stress conditions due to drought [24,25]; in addition, Santos-Torres et al. [20] reported AP21’s capacity to solubilize Pi and T88’s ability to mineralize organic P (Po). Likewise, Pardo-Díaz et al. [26] found that AP21 and D7 showed potential in reducing nitrogen (N) fertilization. In this context, the use of organic and inorganic P sources in combination with PSB has the potential to contribute to increasing soil P availability and improving pasture productivity. Therefore, the objective of the present study was to evaluate the effect of PSB inoculation and the application of P sources on soil P fractions and the productivity of Kikuyo grass.

## 2. Materials and Methods

### 2.1. Description of the Experimental Site

The study was conducted at the Tibaitatá Research Center of the Colombian Corporation for Agricultural Research (AGROSAVIA), located in Mosquera, Cundinamarca, in the Colombian high tropic region (4°41${}^{\prime }$43.13${}^{\prime \prime }$ N; 74°12${}^{\prime }$18.77${}^{\prime \prime }$ W) (Figure 1a). The site is located at an altitude of 2600 m a.s.l. During the field experiment period (September 2020–March 2022), the rainfall was in the range of 4.2–165.6 mm month${}^{-1}$, and the air temperature ranged from 0 to 24.4 °C month${}^{-1}$ (Figure 1c).

The experiment was established on a 340 m${}^{2}$ area of Kikuyu grass (*Cenchrus clandestinus* [Hochst. ex Chiov] Morrone) with low annual productivity (10 ton ha${}^{-1}$ dry matter). The soil was classified as Andisol (Typic Hapludands), with phosphate retention up to 85% [27]. The physicochemical characteristics of the soil are shown in Table 1.

The experimental area was prepared by cross-passes of a mower brush cutter and the subsequent restoration of the grassland process with a rigid chisel plow. After 45 days, eight treatments were established (four P sources * two levels of PSB inoculation) with three replicates. The following four P sources were used: (i) control, (ii) rock phosphate (RP), (iii) diammonium phosphate (DAP), and (iv) compost (OM). Two inoculation levels were used: (i) with PSB inoculation and (ii) without PSB inoculation. Treatments were established in 24 plots of 3 × 2 m using a completely randomized design (Figure 1b), and the experiment was set up for 18 months (September 2020–March 2022).

The P sources were designed according to their different levels of solubility. RP, which was gathered from subway deposits in the state of Boyacá-Colombia (Phosphorite), was applied as a source of low-solubility Pi (11.4% P). OM, which was derived from the compost of bovine manure and pruning, was defined as a source of low-solubility Po (0.56% P). Finally, DAP was defined as a source of high-solubility P (20.1% P). The P sources were applied at an annual rate of 35 kg ha${}^{-1}$ of P as follows: RP (134.5 kg ha${}^{-1}$), DAP (76 kg ha${}^{-1}$), and OM (2729.2 kg ha${}^{-1}$). Finally, an annual dose of N (33 kg ha${}^{-1}$) was applied.

The following consortium of bacterial strains was used: (i) *Herbaspirillum* sp. AP21, (ii) *Azospirillum brasilense* D7, and (iii) *Rhizobium leguminosarum* T88. These strains were gathered from the germplasm bank of microorganisms of AGROSAVIA, Colombia. AP21 and D7 were isolated from Kikuyu grass, while T88, a symbiotic nitrogen-fixing bacterium, was isolated from nodules of red clover (*Trifolium pratense*). We selected these strains because previous research has demonstrated their ability to promote red clover and ryegrass growth under abiotic stress (drought) conditions [24,25] and to optimize P and N use [20,26]. The guaranteed concentration of each strain was 1 × 10${}^{9}$ CFU for strains AP21, T88, and D7.

Inoculation by PSB was carried out after each productive cutting (when the plant reached the 4.5 true leaf stage) with a dose of 2000 mL ha${}^{-1}$ of each inoculant strain in the respective treatments, mixing the strains at the time of application.

### 2.2. Soil P Fractions

To evaluate the edaphic P dynamics, soil samples were collected at three points in time: (i) characterization sampling prior to the establishment of the experiment (0 months), (ii) after 6 months (6 months), and (iii) after 18 months from the establishment of the experiment (18 months) (Figure 1).

Initially, the determination of total P was carried out using the method developed by Murphy and Riley [28]. Then, using the sequential fractionation methodology proposed by Gatiboni and Condron [29], the determination of five P fractions was performed: (i) soil solution P extracted by CaCl${}_{2}$ 0.01 M (P${}_{Solution}$), (ii) plant available P reserve by Mehlich 3 (P${}_{Mehlich-3}$), (iii) P extracted by NaOH 0.5 M into inorganic P (Pi${}_{NaOH}$) together with (iv) organic P (Po${}_{NaOH}$), and (v) P in primary minerals extracted by HCl 1 M (P${}_{HCl}$). For P quantification, the Pi${}_{NaOH}$ extraction was performed by the technique developed by Dick and Tabatabai [30] for alkaline extracts, whereas the method used by Murphy and Riley [28] was applied to all other acid extracts. Finally, occluded P was estimated as the difference between total P content and the sum of the five extractions already described (P${}_{Occluded}$). To better understand the P dynamics, the fractions were grouped into four major P fractions: (i) labile P (P${}_{Solution}$ + P${}_{Mehlich-3}$), (ii) moderately labile P (Pi${}_{NaOH}$ + P${}_{HCl}$), (iii) occluded P (P${}_{Occluded}$), and (iv) organic P (Po${}_{NaOH}$).

### 2.3. Enzymatic Activity

The enzymatic activity of acid phosphomonoesterase was determined by collecting 50 g samples of moist soil at the initial point at which the experiment was established, as well as 6 and 18 months thereafter. The determination was achieved following the protocol proposed by Tabatabai and Bremner [31] and modified by Tabatabai [32]. To avoid the effect of dissolved organic matter on the determination of phosphomonoesterase activity, soil without substrate was used as a control. Additionally, to avoid the interference of substrate hydrolysis, a control was utilized in which no soil was added [33]. For this, 1 g of soil was added to a 50 mL tube with 0.2 mL of toluene and 4 mL of buffer solution adjusted to pH 6.5 for acid phosphomonoesterases. Then, 1 mL of sodium ρ-nitrophenyl phosphate was added to initiate the reaction, after which it was incubated at 37 °C for 1 h. The reaction was stopped by adding 1 mL of 0.5 M CaCl${}_{2}$ and 4 mL of 0.5 M NaOH. Finally, the obtained suspension was filtered, and the color intensity was measured at 420 nm in a spectrophotometer. The calibration curve was calculated out using ρ-nitrophenol as the standard.

### 2.4. Productive Parameters

For dry matter accumulation, 11 samples were collected between September 2020 (initial or 0 month sampling) and March 2022 (18 month sampling). Each sample was obtained when more than 50% of the plants reached 4.5 true leaves [34]. Briefly, using a 0.25 m${}^{2}$ quadrat in each plot, foliage was cut, weighed, and dried at 60 °C for 48 h to estimate dry matter (kg ha${}^{-1}$ day${}^{-1}$).

The determination of protein and foliar P was carried out in the 0, 6, and 18 month samplings. This determination was achieved using the near-infrared reflectance spectroscopy (NIRS) technique, according to the equation generated for Kikuyu grass by Ariza et al. [35]. The yield per day for each treatment was calculated by multiplying the value given for protein and foliar P by the dry matter value (kg ha${}^{-1}$ day${}^{-1}$).

### 2.5. Statistical Analysis

The soil properties (soil P fractions and acid phosphomonoesterase activity) and production parameters (dry matter, protein, and foliar P) were adjusted using linear mixed effect models (lmer), where the interactions between P sources and inoculation levels (P source * inoculation level) were used as the fixed effects and replications as the random effects, and normality assumptions were checked by exploratory residual analysis. Following the model adjustment, Tukey’s HSD test (*p*-value < 0.05) was applied for differences found among the interactions. Changes in the fractions assessed between uninoculated and inoculated samples were analyzed using Student’s *t*-test. In the multivariate analysis, a Pearson’s correlation coefficient (*p* < 0.05) was applied for correlations found between soil properties and production variables at each time point after establishment and inoculation. All analyses were conducted using R version 4.2.0 [36] and RStudio version 1.3.1 [37].

## 3. Results

### 3.1. Dynamics of P Fractions

Independent of treatment, P was represented in greater proportions by P${}_{Occluded}$ (41–56% of total P). Meanwhile, organic P (Po${}_{NaOH}$) and Pi${}_{NaOH}$ represented average values (13–33% and 16–28%, respectively), whereas labile P (P${}_{Solution}$ and P${}_{Mehlich-3}$) demonstrated low values (4–10%). Thus, 93.4% of the edaphic P was retained under structural components of minerals (P${}_{Occluded}$), adsorbed as moderately labile forms of Fe and Ca phosphates (Pi${}_{NaOH}$ and P${}_{HCl}$), and immobilized as organic forms of P (Po${}_{NaOH}$).

The P${}_{HCl}$ fraction (P associated with Ca complexes) did not change during the evaluation period (Figure 2). At 18 months, inoculation increased the P${}_{HCl}$ in treatments fertilized with OM (Table A1). Pi${}_{NaOH}$ (P associated with Fe complexes) decreased (from 17 to 14.3%) between 6 and 18 months (Figure 2). However, at 18 months, inoculation influenced Pi${}_{NaOH}$ accumulation when DAP was applied (Table A1). The dynamic of P${}_{Occluded}$ (P precipitated in soil minerals) showed a strong reduction between 6 and 18 months (from 53 to 43% of total P) when inoculated (Figure 2). At 18 months, inoculation affected (*p* < 0.05) the accumulation of P${}_{Occluded}$ in the control treatment (Table A1). The organic fraction (Po${}_{NaOH}$) exhibited a strong increase between 6 and 18 months, from 14.7% to 29.31% in relation to total P, with inoculation. However, a lower accumulation was evidenced in treatments inoculated with PSB (29%) with respect to uninoculated treatments (32.9%), especially in those fertilized with RP (Figure 2).

To understand the P lability, we grouped the fractions into four major P fractions: (i) labile P (P${}_{Solution}$ + P${}_{Mehlich-3}$), (ii) moderately labile P (Pi${}_{NaOH}$ 0.5 M + P${}_{HCl}$), (iii) P${}_{Occluded}$, and (iv) organic P (Po${}_{NaOH}$ 0.5 M) (Figure 3a,b). Inoculation with PSB influenced the concentrations of labile P at 6 (*p* < 0.01) and 18 months (*p* < 0.001) and moderately labile P (Mod-Labile P) at 18 months (*p* < 0.05), although total P did not show increased concentrations (850 mg kg${}^{-1}$) between 6 and 18 months.

The labile P fraction was directly related to plant nutrition and productivity. P${}_{Solution}$ refers to the P most available for plant absorption, while P${}_{Mehlich-3}$ refers to the reserve of the labile P fraction that was slightly adsorbed on the surface of the soil colloids.

At 6 months, inoculation had significant effects on the availability of P${}_{Solution}$ in the control treatment (35 mg kg${}^{-1}$) and those fertilized with RP (36.4 mg kg${}^{-1}$), which represented an increase of 37.4 and 44.4%, respectively, with respect to the uninoculated treatment (Figure 3c). Meanwhile, at 18 months, inoculation influenced the control treatment (34.3 mg kg${}^{-1}$) and those fertilized with OM (34.9 mg kg${}^{-1}$) (Figure 3d). For P${}_{Mehlich-3}$, at 6 months, inoculation had significant effects (*p* < 0.05) on treatments fertilized with OM (492.7 mg kg${}^{-1}$), representing a 29.3% increase in available P with respect to the uninoculated treatment (Figure 3e). Moreover, at 18 months, inoculation influenced P${}_{Mehlich-3}$ availability in treatments fertilized with RP (360.2 mg kg${}^{-1}$) and DAP (332.3 mg kg${}^{-1}$) and the ontrol (320 mg kg${}^{-1}$), representing an increase of 88, 79.4, and 113.2%; respectively, with respect to their uninoculated counterparts (Figure 3f).

### 3.2. Phosphomonoesterase Activity in Response to PSB Inoculation and P Sources

Acid phosphomonoesterases are enzymes that hydrolyze Po to Pi in the form of orthophosphoric acid (${}^{-2}$ and ${}^{-1}$) and predominate in acid soils. Inoculation was observed to affect acid phosphomonoesterase activity (Figure 4). Inoculation with PSB independent of P sources increased the activity independent of the time of establisment, although the highest enzyme activity occurred at 6 months (Figure 4a).

At 6 and 18 months, inoculation had effects on enzyme activity in all P sources evaluated (Figure 4a,b). At 6 months, the highest acid phosphomonoesterase activity was recorded with the control and RP-fertilized treatments, with increases of 104.7% and 77.7% occurring due to the effect of the inoculation, respectively (Figure 4a). At 18 months, there was a reduction in activity for all the P sources evaluated, with the exception of fertilization with RP (Figure 4b), which increased and presented the highest activity (244.2 µg ρ-NP g${}^{-1}$ h${}^{-1}$), representing a 63.9% increase with respect to the corresponding uninoculated treatment. Fertilization with DAP as a source of high P solubility resulted in lower phosphomonoesterase activity with respect to the low-solubility P sources.

### 3.3. Productivity and Quality of Kikuyu Grass

The productive response parameters of the Kikuyu grass, such as dry matter, protein, and foliar P accumulation, showed significant differences (*p* < 0.05) in relation to PSB inoculation and P source fertilization. These results suggested that both PSB inoculation and P sources had a significant impact on Kikuyu grass productivity and quality (Figure 5).

At 6 months (Figure 5a,c,e), the effects of inoculation in the control and RP treatments were observed on the dry matter (27.8 and 27.3 kg ha${}^{-1}$ day${}^{-1}$, respectively); protein (4.7 and 4.5 kg ha${}^{-1}$ day${}^{-1}$, respectively); and foliar (0.46 and 0.41 kg ha${}^{-1}$ day${}^{-1}$, respectively) P accumulation. However, concerning protein accumulation, effects after OM fertilization were also observed (4.5 kg ha${}^{-1}$ day${}^{-1}$).

At 18 months (Figure 5b,d,f), inoculation influenced dry matter, protein, and foliar P accumulation for the RP, DAP, and control treatments. However, the highest accumulations in terms of Kikuyu productive parameters were observed in the treatment fertilized with RP (46.7 kg dry matter ha${}^{-1}$ day${}^{-1}$, 7.52 kg protein ha${}^{-1}$ day${}^{-1}$, and 1.12 kg foliar P ha${}^{-1}$ day${}^{-1}$, respectively) with increases in accumulation of 45.2, 40.1, and 54.2%, respectively, in relation to the uninoculated treatment. The highest accumulations in the productive parameters of Kikuyu grass occurred at 18 months.

Additionally, the correlation between soil properties (P fractions and acid phosphomonoesterase activity) and the productive parameters of Kikuyu grass was determined by analyzing the effect of PSB inoculation at 6 and 18 months after experiment establishment (Figure 6).

At 6 months after experiment establishment, there were significant and positive correlations between the accumulation of productive variables and P${}_{Solution}$ and acid phosphomonoesterase activity in the uninoculated treatment (Figure 6a). In the inoculated treatment, there were significant negative correlations between dry matter accumulation and the P${}_{Mehlich-3}$ fraction and positive correlations between acid phosphomonoesterase and protein accumulation (Figure 6c).

Likewise, at 18 months, there were significant and positive correlations between the accumulation of dry matter, protein, and foliar P and P${}_{Mehlich-3}$ and acid phosphomonoesterase activity in the uninoculated treatment (Figure 6b). The inoculated treatments showed larger correlation networks than the other treatments, and the productive parameters showed a significant and positive correlation with the P${}_{Mehlich-3}$, P${}_{HCl}$, and protein accumulation. Moreover, there was a positive correlation between acid phosphomonoesterase activity and P${}_{Solution}$ (Figure 6d).

In general, the results demonstrated significant improvements in the availability of soil labile P and the productivity of Kikuyu grass with PSB inoculation. In addition, a strong positive correlation between acid phosphomonoesterase activity and dry matter, protein, and foliar P accumulation was found. These results could indicate an important biotechnological tool to reduce the application of synthetic P sources, increasing the dry matter accumulation and nutritional quality of Kikuyu grass.

## 4. Discussion

### 4.1. Soil P Dynamics

The low availability of P in Andisol has prompted the search for strategies to improve the availability of this nutrient. In response to this problem, this study demonstrated the potential of PSB inoculation with different P sources to improve soil P lability, thus transforming insoluble forms of P into usable forms for Kikuyu grass, and in turn reducing the application of synthetic fertilizers. Preliminary studies have shown that PSB inoculants improve soil P dynamics and promote soil P legacy circularity [25,26,38].

PSB enhances the capacity of indirect P solubilization by the production of organic acids resulting from bacterial metabolism that chelate P-bound cations and compete with P for adsorption sites [39,40,41], thus optimizing soil P use [42]. Regarding the bacterial strains evaluated in this study, Santos-Torres et al. [20] demonstrated that *Herbaspirillum* sp. AP21 and *Rhizobium legominusarum* T88, in culture medium supplemented with RP, have the capacity to produce the following organic acids 12 days after inoculation: oxalic, gluconic, formic, lactic, and acetic.

Additionally, the high root density of grasses increases P uptake with low-solubility sources, such as RP, due to the continuous solubilization of P in the soil solution [20,43,44].

Inoculation had a positive influence on soil P dynamics, mainly on P${}_{Solution}$ and P${}_{Mehlich-3}$ in the treatment fertilized with RP (Figure 2). The P${}_{Mehlich-3}$ fraction represented the greatest influence according to its soil concentrations, which exceeded those of P${}_{Solution}$ by up to 10 times, showing that P${}_{Mehlich-3}$ is a P reservoir fraction in the soil that is mobilized according to plant uptake [45].

Reductions in Pi fractions (P${}_{Occluded}$, P${}_{Mehlich-3}$, and Pi${}_{NaOH}$) contributed to an increase in Po (Po${}_{NaOH}$) concentration (Figure 2). It was inferred that the desorption and dissolution of Pi occurred in colloids and secondary minerals, forming organo-mineral cation–metal (Fe/Al) complexes with soil humic substances [46], increasing Po [47,48], due to the type of soil evaluated and its high OM contents [9,49,50], as well as the use of the soil in Kikuyu grasslands as ground cover [10,51,52].

Tiecher et al. [47], found reductions in the proportion of Pi (from 57 to 49%) and consequently an increase in the proportion of Po (from 43 to 51%) in terms of total P, evaluating, for 23 years, no-tillage systems in cover crops with annual fertilization. Similarly, Dos Santos et al. [48], reported increases in the soil content of total P and OM in cover cropping systems, which contributed to the transformation of Pi into Po, reflected in a low rate of organic carbon mineralization due to the absence of soil management.

As noted in previous studies, native and perennial grasses have the ability to maintain a higher proportion of moderately available Po over time [53,54], especially in C${}_{4}$ perennial grasses, such as Kikuyu grass, that have the ability to store more carbon because of their high root biomass [55], thus demonstrating the important role of Po mineralization in P availability for grasses [56].

Additionally, the PSB inoculation influenced (*p* < 0.05) the soil chemical parameters evaluated, as shown in Table A2. The highest pH values were found in the treatments fertilized with OM and RP at 6 and 18 months after the establishment of the experiment. The pH change could be related to the decreases in the occluded and Fe-associated P fractions (P${}_{occluded}$ and Pi${}_{NaOH}$) observed at 18 months. A pH increase could improve the desorption of P from Fe–P complexes [44]. OM mineralization with PSB inoculation [57] increased soil cation availability (Ca${}^{++}$, Mg${}^{++}$, and K${}^{+}$) (Table A2). Santos-Torres et al. [20] demonstrated the capacity of inoculated strains to produce enzymes (phosphomonoesterases and phytases) related to P-organic mineralization.

### 4.2. Acid Phosphomonoesterase Activity

Acid phosphomonoesterase activity is a response by plants and soil organisms to P deficiency, facilitating Po mineralization, which increases its availability [18,58,59,60]. However, there was high acid phosphomonoesterase activity at 6 and 18 months despite the high labile P contents (>90 mg kg${}^{-1}$), in agreement with the findings of Bush et al. [61] that inoculation with PSB increases plant available P because phosphomonoesterases have a greater influence on Po mineralization, independent of Pi deficiency [60].

Treatments with high-solubility sources (DAP) generated low acid phosphomonoesterase activity, because the application of these Pi sources could repress the synthesis of these enzymes in the soil [58]; meanwhile, RP-fertilized treatments presented a positive effect with PSB inoculation, which was in agreement with the results of Redel et al. [14], emphasizing the close relationship between phosphomonoesterase activity and P uptake in RP-fertilized Andisol.

The results indicated that acid phosphatase activity influences the availability of P${}_{Mehlich-3}$, the P pool in the labile fraction, which is also considered readily available to plants. According to Cross and Schlesinger [62], an increase in the labile P pool is likely the result of Po mineralization, a product of the metabolic activity of bacteria. Enzymatic activity is also a measure of the potential contributions of Po to the supply of orthophosphate or available P and has direct implications for increasing tropical pasture production on Andisol [14,63,64].

### 4.3. Productive Parameters of Kikuyu Grass

Dry matter accumulation was higher at 18 months in treatments with PSB inoculation fertilized with RP. These results coincided with the higher acid phosphomonoesterase activity (Figure 5) and the higher availability of P${}_{Solution}$ and P${}_{Mehlich-3}$ in the labile fraction of the soil. Beltran et al. [38] found a greater availability of P in the labile fractions with the inoculation of *Rhizobium* sp., improving growth and development in annual forage crops and P nutrition, in turn allowing a reduction in the application of mineral P fertilizers to 50%. The synergism between PSB inoculation and low-solubility P sources increased dry matter accumulation in Kikuyu grass relative to high-solubility mineral sources. In grasses, the increase in dry matter availability is related to leaf growth in the short region that continuously produces cells, located near the point of attachment of the leaf to the base of the stem [65]. Thus, an adequate supply of P increases the rates of cell division and expansion [66]; therefore, it also increases biomass accumulation. Regarding the availability of protein and foliar P, the highest accumulations were presented by PSB inoculation in treatments fertilized with RP. From these results, it could be inferred that the increases in protein and foliar P accumulation found in this study were possibly due to the greater acquisition of N in plants due to the effects of biological fixation by the inoculated strains [26] and the greater availability of P, since the synergistic effect of the activity of the inoculated strains stimulated growth and improved N and P uptake by the plant [67]. Estrada et al. [68] mentioned that PSB inoculation is important for soil P adsorption and improves N use efficiency and its content in aerial tissues.

Santos-Torres et al. [20] found that inoculation with *Rhizobium legominusarium* T88 and *Herbaspirillum* sp. AP21 in Ryegrass and Red Clover produced higher foliar P contents along with the application of DAP and RP in doses of 75-25%, respectively. Khan et al. [42] also reported 15% increases in fresh biomass and significant correlations between soil and foliar P availability in Mungo bean under field conditions and inoculation with *Bacillus mageterium* and *Bacillus polymyxa* in joint application with RP, which agreed with the findings evidenced in this study.

## 5. Conclusions

This study indicated that the inoculation of *Herbaspirillum* sp. AP21, *Azospirillum brasilense* D7, and *Rhizobium legominusarium* T88 improved soil P dynamics, especially the availability of P in the labile fraction (P${}_{Solution}$ and P${}_{Mehlich-3}$). Fertilization with OM increased labile P (P${}_{Mehlich-3}$), acting as an important P reservoir that facilitated its supply depending on the plant’s demand, whereas fertilization with a low-solubility inorganic source (RP) increased the availability of P in both the soil solution and its reserve (P${}_{Mehlich-3}$). These low-solubility P sources together with PSB inoculation led to higher acid phosphomonoesterase activity and therefore presented a potential technology for increasing the productivity and forage quality of Kikuyu grass (a grass with high P fertilization requirements) as a consequence of the improvement in P availability in the labile fraction of the soil. However, further studies conducted under field conditions, specifically tracking the strains in the plant tissue and rhizospheric and bulk soil, are necessary to establish the stability of this technology.

## Figures and Tables

**Figure 1 microorganisms-11-01748-f001:**
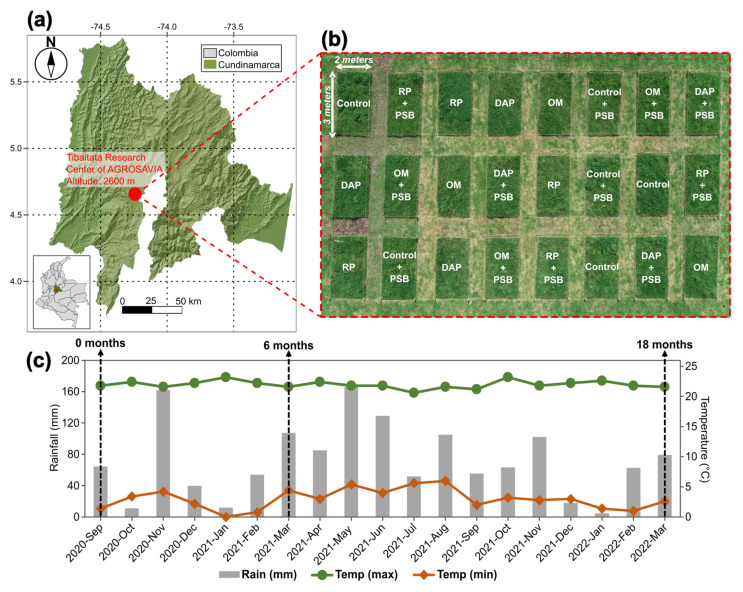
Geographic location of the experiment. (**a**) Map of Cundinamarca describing the location of the experiment. (**b**) Experimental design. (**c**) Temperature (maximum and minimum) and rainfall during time after experiment establishment.

**Figure 2 microorganisms-11-01748-f002:**
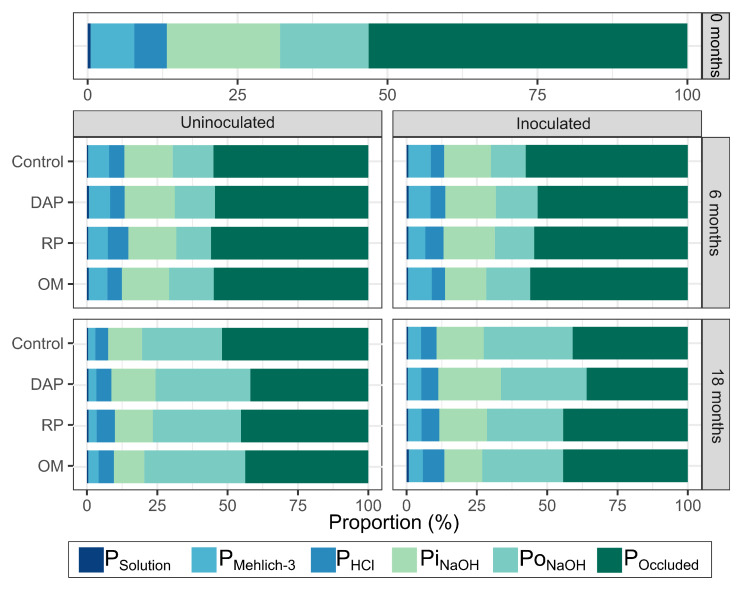
Soil P proportions of inoculated and uninoculated treatments from 0 to 6 and 18 months after experiment establishment.

**Figure 3 microorganisms-11-01748-f003:**
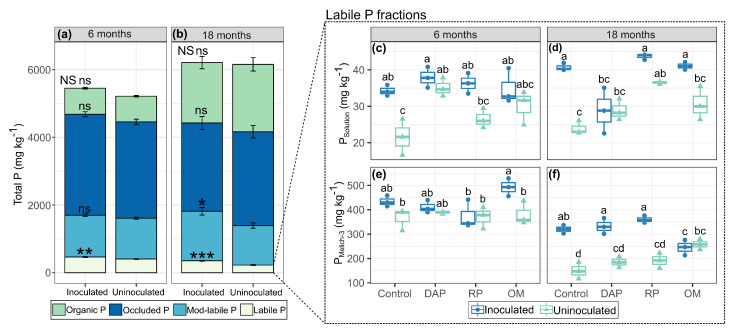
Mean soil P concentration for inoculated and uninoculated treatments. (**a**) 6 months and (**b**) 18 months, grouped into P fractions: organic P (Po${}_{NaOH}$), P${}_{Occluded}$, moderately labile P (Mod-Labile P), and labile P. Labile P fractions P${}_{Solution}$ (**c**) and P${}_{Mehlich-3}$ (**e**) 6 months and (**d**,**f**) 18 months after experiment establishment. Error bars indicate standard error. Asterisks indicate that mean values differed significantly from each other at the *p* < 0.05 (*), *p* < 0.01 (**), *p* < 0.001 (***), and ns (not significant) levels for each time point during the experimental period according to Student’s *t*-test. Means followed by the same letters did not differ significantly according to Tukey’s test (*p*-value < 0.05).

**Figure 4 microorganisms-11-01748-f004:**
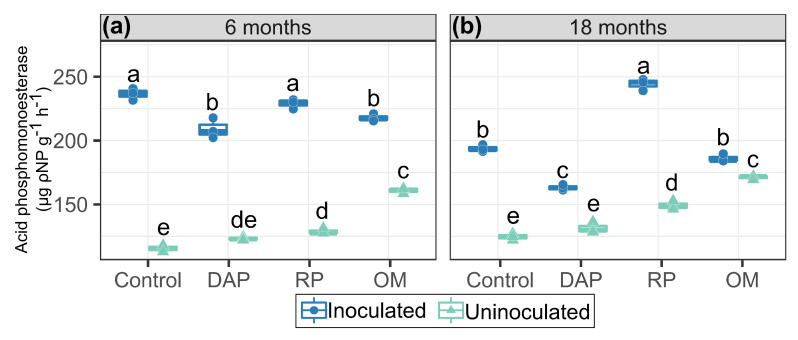
Enzymatic activity of acid phosphomonoesterases. (**a**) 6 months and (**b**) 18 months after experiment establishment. Means followed by the same letters did not differ significantly according to Tukey’s test (*p*-value < 0.05).

**Figure 5 microorganisms-11-01748-f005:**
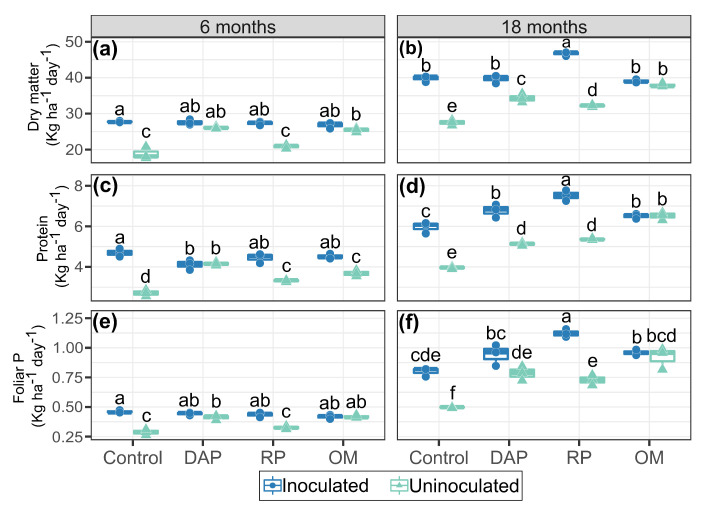
Production and quality of Kikuyu grass (**a**,**c**,**e**) 6 months and (**b**,**d**,**f**) 18 months after experiment establishment. Means followed by the same letter did not differ according to Tukey’s test (*p* < 0.05).

**Figure 6 microorganisms-11-01748-f006:**
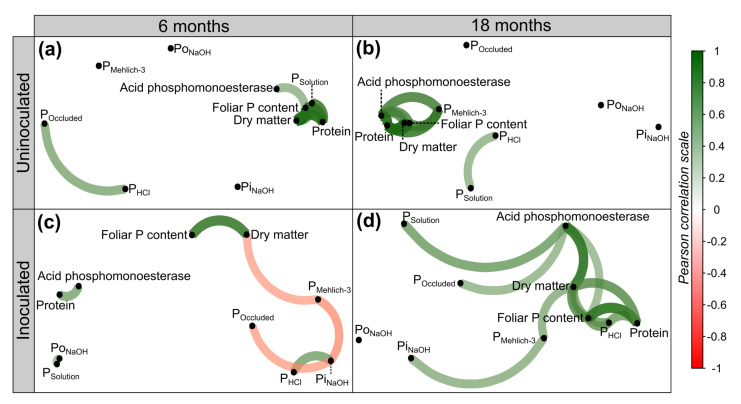
A significant (*p*-value < 0.05) Pearson’s correlation network between soil properties and the dry matter, protein, and foliar P accumulation of Kikuyu grass. Inoculated and uninoculated treatments at (**a**,**c**) 6 months and (**b**,**d**) 18 months after establishment. Green indicates a positive correlation, and red indicates a negative correlation.

**Table 1 microorganisms-11-01748-t001:** Soil physicochemical properties of the experimental site.

Sand	Silt	Clay	H	Bulk Density	OM	K	S	Ca	Mg
(%)	(%)	(%)		$\left(g\;{\mathrm{cm}}^{-3}\right)$	$\left(g\;\;{\mathrm{kg}}^{-1}\right)$	(cmol ${\mathrm{kg}}^{-1}$ )	$\left(\mathrm{mg}\;{\mathrm{kg}}^{-1}\right)$	$\left(\mathrm{cmol}\;{\mathrm{kg}}^{-1}\right)$	$\left(\mathrm{cmol}\;{\mathrm{kg}}^{-1}\right)$
32.9	433	23.8	60	0.44	11.2	4.2	27.4	18	7.8

## Data Availability

Data are contained within the article.

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
