# Peer review of "Phosphate-Solubilizing Bacteria with Low-Solubility Fertilizer Improve Soil P Availability and Yield of Kikuyu Grass"

_microorganisms, 2023, doi:10.3390/microorganisms11071748_

Round 1

Reviewer 1 Report

This manuscript describes the application of Herbaspirillum sp. AP21, Azospirillum brasilense. D7 and Rhizobium legominusarium as phosphate-solubilizing bacteria can improve soil P availability and yield of Kikuyu grass. Dynamics of P fractions, phosphomonoesterase activity and productivity and quality of Kikuyu grass were determined. The novelty of this study is relatively insufficient because it is well-known that phosphate-solubilizing bacteria can promote soil P availability. However, there are still contributions in field experiments for Kikuyu grass. There are some issues that need to be addressed in the manuscript.

1. Is there detection of dynamic changes in soil pH value? Soil pH will affect soluble P.

2. Have you analyzed the changes in these microbial communities including Herbaspirillum sp. AP21, Azospirillum brasilense. D7 and Rhizobium legominusarium?

3. Organic acids produced by phosphate-solubilizing bacteria also can improve soil P availability. Has soil organic acid been analyzed?

4. Has the soil physicochemical properties such as Table 1 including soil elements, organic matter and total nitrogen been analyzed during 18 months? The dynamic change of soil physicochemical properties can explore the relationship between the uninoculated and inoculated treatment.

Author Response

Point 1. This manuscript describes the application of Herbaspirillum sp. AP21, Azospirillum brasilense. D7 and Rhizobium legominusarium as phosphate-solubilizing bacteria can improve soil P availability and yield of Kikuyu grass. Dynamics of P fractions, phosphomonoesterase activity and productivity and quality of Kikuyu grass were determined.

Response 1: We appreciate the thoughtful analysis and careful reading of the manuscript. We have adjusted the manuscript to include most of the suggestions. We think these have greatly improved our manuscript.

Point 2. The novelty of this study is relatively insufficient because it is well-known that phosphate-solubilizing bacteria can promote soil P availability. However, there are still contributions in field experiments for Kikuyu grass. There are some issues that need to be addressed in the manuscript.

Response 2: We agree with your opinion, numerous studies have focused on the influence of inoculation both on plant development and the total P accumulation in the soil under controlled conditions. However, this study is novel because it demonstrated that PSB inoculation increases the availability of soil P under field conditions, which remains largely unexplored. This question is complex, context-specific, and needs addressing to uncover new insights into the effectiveness of PSB. Here, soil P fractionation was performed 6- and 18-months post-establishment of the experiment, assessing the P dynamics. We demonstrated that the PSB inoculation increases the soil labile P and is related to a progressively improving to yield, grass quality, and soil functionality (Phosphomonoesterase activity).

Point 3. 1. Is there detection of dynamic changes in soil pH value? Soil pH will affect soluble P.

Response 3: We agree with your suggestion. Indeed, the dynamics of changes in pH values in the different treatments were monitored. Thus, Table A2 has been included in the Appendix section, with information on the soil chemical parameters evaluated during the experimental period. Additionally, we added a paragraph within Discussion section. Please see lines 279 - 287: "Additionally, the PSB inoculation influenced (p < 0.05) the soil chemical parameters evaluated, as shown in Table A2. The highest pH values were found in the treatments fertilized with OM and RP at 6- and 18- months after establishment of the experiment. The pH change could be related to decreases in the occluded and Fe-associated P fraction (Pi NaOH and P occluded) observed at 18 months. The pH increase potentially improves the desorption of P from Fe-P complexes (Biswas et al., 2022). The OM mineralization with PSB inoculation increases soil cations availability (Ca++, Mg++ and K+). Santos-Torres et al. 2021 demonstrated the capacity of inoculated strains to produce enzymes (Phosphomonoesterases and Phytases) related with P-organic mineralization”.

Point 4. 2. Have you analyzed the changes in these microbial communities including Herbaspirillum sp. AP21, Azospirillum brasilense. D7 and Rhizobium legominusarium?

Response 4: Unfortunately, in the present work, we did not specifically monitor changes in soil microbial community and presence of the inoculated microbes in the rhizosphere or the plant tissue. In fact, we are currently developing molecular biology methods for tracking of these strains using quantitative PCR and GFP-based reporters. Thanks for your suggestion. We added a sentence with this topic as future research to develop in conclusions section. Please see line 347 – 349, “However, further studies conducted under field conditions, specifically tracking the strains in the plant tissue and rhizospheric and bulk soil, are necessary to establish the stability of the technology”.

Point 5. 3. Organic acids produced by phosphate-solubilizing bacteria also can improve soil P availability. Has soil organic acid been analyzed?

Response 5: Thanks for your suggestion. We added a paragraph in Discussion section where we performed physiological and genomic characterization of used strains. Please see lines 250 to 254: “Regarding the bacterial strains evaluated in this study, Santos - Torres et al. (2021) demonstrated that Herbaspirillum sp. AP21 and Rhizobium legominusarium. T88, in culture medium supplemented with RP, have the capacity to produce the following organic acids: oxalic, gluconic, formic, lactic and acetic, 12 days after inoculation”.

Point 6. 4. Has the soil physicochemical properties such as Table 1 including soil elements, organic matter and total nitrogen been analyzed during 18 months? The dynamic change of soil physicochemical properties can explore the relationship between the uninoculated and inoculated treatment.

Response 6: As mentioned in point 3, we included Table A2 in the Appendix section, with information on the main soil chemical parameters evaluated, which includes all parameters listed.

Reviewer 2 Report

The manuscript entitled « Phosphate-solubilizing bacteria with low-solubility fertilizer

improves soil P availability and yield of Kikuyu grass» is devoted the productive response of Kikuyu grass and soil P dynamics evaluation after different P sources inoculation.

The topiс is important and will be of interest to a wide range of researchers in pedology and agriculture science. The data obtained in this study have a high biotechnological potential in improving P uptake by plants. It is important to note that the data were obtained by the authors as a result of laboratory and field experiments on a 340 m2 area of Kikuyu grass.

In general, I liked the design of the experiment and the way the results were presented. The authors obtained important agrotechnical results showing the effectiveness of the consortium of strains they proposed. The only thing I have doubts about the correspondence of the material to the goals and objectives of the journal Microorganisms. If the authors could add data on changes in the microbial composition of soils during the experiment, this would greatly decorate the work. On the whole, I can recommend the manuscript for publication in this form.

Author Response

Point 1. The manuscript entitled « Phosphate-solubilizing bacteria with low-solubility fertilizer improves soil P availability and yield of Kikuyu grass» is devoted the productive response of Kikuyu grass and soil P dynamics evaluation after different P sources inoculation. The topiс is important and will be of interest to a wide range of researchers in pedology and agriculture science. The data obtained in this study have a high biotechnological potential in improving P uptake by plants. It is important to note that the data were obtained by the authors as a result of laboratory and field experiments on a 340 m2 area of Kikuyu grass.

Response 1: We are grateful for your review and for considering our research to be appropriate and relevant.

Point 2. In general, I liked the design of the experiment and the way the results were presented. The authors obtained important agrotechnical results showing the effectiveness of the consortium of strains they proposed. The only thing I have doubts about the correspondence of the material to the goals and objectives of the journal Microorganisms. If the authors could add data on changes in the microbial composition of soils during the experiment, this would greatly decorate the work. On the whole, I can recommend the manuscript for publication in this form.

Response 2: Unfortunately, in the present work, we did not specifically monitor changes in soil microbial community and presence of the inoculated microbes in the rhizosphere or the plant tissue. In fact, we are currently developing molecular biology methods for tracking of these strains using quantitative PCR and GFP-based reporters. Thanks for your suggestion. We added a sentence with this topic as future research to develop in conclusions section. Please see line 347 – 349, “However, further studies conducted under field conditions, specifically tracking the strains in the plant tissue and rhizospheric and bulk soil, are necessary to establish the stability of the technology”.

Round 2

Reviewer 1 Report

I agree to accept the revision.